

# Variability in sea ice carbonate chemistry: A case study comparing the importance of ikaite precipitation, bottom ice algae, and currents across an invisible polynya

Brent G. T. Else[1], Araleigh Cranch[1], Richard P. Sims[1,2], Samantha Jones[1], Laura A. Dalman[3,4], Christopher J. Mundy[3], Rebecca A. Segal[5,6], Randall K. Scharien[7], Tania Guha[1]

[1]Department of Geography, University of Calgary, Calgary, Alberta, Canada

[2]Now at: College of Life and Environmental Sciences, University of Exeter, Exeter, United Kingdom

[3]Centre for Earth Observation Science, University of Manitoba, Winnipeg, Manitoba, Canada

[4]Now at: Institute for Marine and Antarctic Studies, University of Tasmania, Hobart, Australia

[5]Arctic Eider Society, Sanikiluaq, Canada

[6]SmartIce Sea Ice Monitoring & Information Inc., St. John's, Canada

[7]Department of Geography, University of Victoria, Victoria, British Columbia, Canada

*Correspondence to*: Brent G. T. Else (belse@ucalgary.ca)





**Abstract.** The carbonate chemistry of sea ice is known to play a role in global carbon cycles, but its importance is uncertain in part due to disparities in reported results. Variability in physical and biological drivers is usually invoked to explain differences between studies. In the Canadian Arctic Archipelago, "invisible polynyas" – areas of strong currents, thin ice, and potentially high biological productivity – are examples of extreme spatial variability. We used an invisible polynya as a natural laboratory to study the

effects of inferred initial ice formation conditions, ice growth rate, and algal biomass on the distribution of carbonate species by collecting enough cores to perform a statistical comparison between sites located within, and just outside of, a polynya near Iqaluktuttiaq (Cambridge Bay, Nunavut, Canada). At both sites, the uppermost 10-cm ice horizon showed evidence of $CO_2$ offgassing, while carbonate distributions in the middle and bottommost 10-cm horizons largely followed the salinity distribution. In the polynya,

the upper-ice horizon had significantly higher bulk total inorganic carbon (TIC), total alkalinity (TA), and salinity, potentially due to freeze-up conditions that favoured frazil ice production. The middle-ice horizons were statistically indistinguishable between sites, suggesting that ice growth rate is not an important factor for the carbonate distribution under mid-winter conditions. The thicker (non-polynya) site experienced higher algal biomass, TIC, and TA in the bottom horizon. Carbonate chemistry in the

bottom horizon could be explained by the salinity distribution, with the strong currents at the polynya site potentially playing a role in desalinisation; biology did not have a noticeable impact. We did see evidence of calcium carbonate precipitation, but with little impact on the TIC:TA ratio, and little difference between sites. Because differences were constrained to relatively thin layers at the top and bottom, vertically averaged values of TIC, TA, and especially the TIC:TA ratio were not meaningfully different between

sites. This provides some justification for using a single bulk value for each parameter when modeling sea ice effects on ocean chemistry at coarse resolution. Exactly what value to use (particularly for the TIC:TA ratio) likely varies by region but could potentially be approximated from knowledge of the source seawater and sea ice salinity. Further insights await a rigorous intercomparison of existing data.





## 1 Introduction

Polar seas typically act as strong sinks for atmospheric carbon dioxide. Recent estimates of the Arctic $CO_2$ sink suggest an uptake on the order of 150-200 TgC year[-1], or about 10-14% of the global ocean sink (Bates and Mathis, 2009; Yasunaka et al., 2016; Manizza et al., 2019). This sink is significantly larger than one might expect given the relatively small surface area (about 3% of the global surface ocean). $CO_2$
uptake for the sea ice zone of the Southern Ocean has been estimated at 56 TgC year[-1] or about 4% of the global sink (Takahashi et al., 2009), slightly exceeding its relative surface area (3%).

The carbon sink in these polar regions is thought to be enhanced by some combination of the solubility pump ($CO_2$ is more soluble in cold water, and thus can be transported to depth by convection events), the biological pump ($CO_2$ sequestered in organic material is exported out of the surface ocean by sinking),
and the "sea ice pump". The sea ice pump was first proposed by Rysgaard et al. (2007) who observed elevated total alkalinity (TA) relative to total inorganic carbon (TIC) in melted sea ice samples. The hypothesized cause of this TA enrichment was the precipitation of ikaite, a calcium carbonate crystal form that has since been positively identified in Antarctic (Dieckmann et al., 2008) and Arctic (Dieckmann et al., 2010) sea ice. The theory posits that as sea ice grows and rejects impurities, ikaite
crystals are formed and preferentially trapped in the sea ice matrix relative to brine. Ikaite is a form of particulate inorganic carbon, but its precipitation lowers the TIC concentration of the source brine by only 1 mol while lowering TA by 2 mol. Brine that drains out of the ice during the desalination process (Notz and Worster, 2009) is therefore relatively enriched in DIC, leaving the bulk sea ice enriched in TA. If DIC-rich brines sink to sufficient depths, this sequesters carbon from the surface ocean, while the
subsequent ice melt in spring releases TA to the surface ocean, lowering $pCO_2$ and driving higher atmospheric $CO_2$ uptake during the open water season.

The fundamental parameter driving this pump is the TA:TIC ratio in sea ice, which if only affected by

ikaite should range from 2:1 (the theoretical maximum ratio) to approximately 1:1 (seawater typically has a ratio near 1:1, which should be reflected in sea ice if brine is rejected conservatively). Published results have reported ratios spanning 0.9 to 2.2 from a variety of different Arctic and Antarctic field sites (Rysgaard et al., 2007, 2009; Miller et al., 2011a, 2011b; Geilfus et al., 2012; Brown et al., 2015; Findlay et al., 2015). The absolute concentration values of TIC and TA in sea ice also span a significant range: for the above listed studies, these values range from <100 µmol kg$^{-1}$ to >700 µmol kg$^{-1}$ for both species. This observed variability suggests not only that ikaite precipitation is a variable process, but that other biogeochemical factors impact TIC and TA in sea ice.

Ice growth conditions are one set of factors that may affect sea ice carbonate chemistry. The physical process of sea ice formation depends in part on turbulence (Martin and Kauffman, 1981; Savel'yev 1958, 1963). During initial ice formation, turbulent areas (driven by wind or tidal currents) generate large amounts of frazil ice in a thicker mixed layer, leading to a collision and bonding process that results in faster formation of floes (Martin and Kauffman, 1981; Chadian and Strickland-Constable, 1974; Chalmers and Williamson, 1965; Savel'yev 1958, 1963). In contrast, calmer areas develop thinner frazil ice layers (Carnat et al., 2013). Once the surface ice (approximately the first 30 cm or less) is consolidated, congelation (or columnar) ice growth dominates, and the rate of growth strongly impacts brine rejection. Faster crystallization of ice during sea ice formation typically results in increased retention of salts, including TIC. A laboratory study by König et al. (2018) found the amount of TIC trapped in sea ice almost doubled for sea ice grown at -40°C compared to ice grown at -15°C. In the natural environment, ice growth rates are controlled not only by atmospheric forcing, but also by snow depth (deep snow insulates the ice and lowers the growth rate) and under ice currents (strong currents enhance ocean heat flux and lower the ice growth rate).

Gas exchange between the sea ice and the overlying atmosphere can also play a role in sea ice carbonate chemistry, at least in the upper layers of the ice. During ice formation, some brine is rejected upwards, allowing $CO_2$ to offgas to the atmosphere (Geilfus et al., 2013; Nomura et al., 2014). In the frazil crystal



structures of surface ice, it is possible that the high brine fraction creates enough permeability for this outgassing to occur over a layer several centimetres thick. Laboratory estimates have found that the amount of carbon released to the atmosphere via this process is a small fraction (<1%) of the total TIC redistributed by ice formation (Rysgaard et al., 2009; König et al, 2018). While gas exchange may not impact the entire ice column, it may be important in determining the vertical distribution of TIC in the surface horizon of the sea ice.

Biological processes are also likely to affect the vertical distribution of TIC and TA in sea ice. Miller et al. (2011a) found evidence for the production of $CO_2$ throughout the sea ice column in winter, likely as a result of respiration within brine inclusions. In the spring, algal blooms occur within the bottom of sea ice in many Arctic locations (Leu et al., 2015) and throughout the ice profile in Antarctic regions (e.g., Meiners et al., 2012, 2018), resulting in observed decreases in TIC in the lower horizons of sea ice cores (Dellile et al., 2007; Miller et al. 2011a; Brown et al., 2015). The spatial distribution of ice algae is known to be highly heterogeneous (Leu et al., 2015), contributing to high heterogeneity in other biogeochemical properties, including carbonate chemistry (Miller et al., 2015). In general, ice algal biomass is controlled by light availability (largely a function of snow cover, with thick snow reducing light transmission and ice algae production, e.g., Campbell et al. (2014) and nutrient availability (potentially a function of under-ice turbulence, with strong currents enhancing nutrient supply and ice algal production, e.g., Dalman et al. (2019).

The horizontal and vertical variability in sea ice TIC and TA that results from these processes may impact the efficiency of the sea ice carbon pump. In an early study, Rysgaard et al. (2011) used a simple box model and estimated its impact on air-sea $CO_2$ uptake to be 31in the Arctic and 52 TgC year$^{-1}$ in the Antarctic. However, subsequent efforts using coupled 3-dimensional general circulation models produced much lower estimates. Grimm et al. (2016) found an enhancement of only 2-14 TgC year$^{-1}$ for the Arctic and Antarctic Sea ice zones combined, while Moreau et al. (2016) attributed only a 4 TgC year$^{-1}$ uptake globally to the sea ice pump. Using a one-dimensional model with more detailed biogeochemical





processes, Mortenson et al. (2018) similarly found that ikaite precipitation and dissolution did not have a large impact on annual air-sea $CO_2$ flux budgets. In all cases, this limited effect was primarily due to the
lack of deep brine convection in winter across most Arctic and Antarctic regions. However, both Grimm et al. (2016) and Moreau et al. (2016) concluded that the sea ice carbon pump is important in certain polar regions (i.e., deep water formation areas), and could be important on long timescales (e.g., during the glacial/interglacial transitions).

In sensitivity analyses, both Mortensen et al. (2018) and Grimm et al. (2016) found that the strength of
the sea ice pump responds strongly to the prescribed TA:TIC ratio. Furthermore, Grimm et al. (2016) found that the absolute concentration of TA and TIC had a significant impact. It is also important to consider that all these studies were working from limited field data. Despite this sensitivity, and the observed variability in sea ice carbonate chemistry described above, the central results of most modeling studies use a single value for TIC, typically around 400 µmol kg$^{-1}$, and a single TA:TIC ratio, typically
2:1 (Rysgaard et al., 2011; Grimm et al., 2016; Mortensen et al., 2018).

If the biogeochemical processes responsible for variability in sea ice carbon system parameters can be constrained by field measurements, that variability could conceivably be included in biogeochemical models, leading to better estimates of the role of the sea ice carbon pump in regional and global models. Accordingly, this paper describes field measurements of TA and TIC in landfast first year sea ice cores
collected at two sites in the Canadian Arctic Archipelago. The sites (separated by only 7 km) were chosen based on prior knowledge of significantly different physical and biological conditions due to the presence of an "invisible polynya" (Dalman et al., 2019) – a region of distinctly thinner ice within the landfast icescape (Melling et al., 2015). This paper seeks to contrast the two sites with objectives to characterize how differing physical and biological processes impact both horizontal and vertical distributions of
carbonate properties in sea ice and determine if these processes affect the bulk estimates of TA and TIC commonly ingested into models of the sea ice carbon pump.



## 2 Methods

### 2.1 Study area

This study was conducted in Dease Strait, a narrow channel that is part of the southern limb of the Northwest Passage in the Canadian Arctic Archipelago. In the winter, Dease Strait is covered by landfast first-year sea ice that typically consolidates in early November, and breaks up in mid-July (Galley et al., 2012, Xu et al., 2021). The strait is relatively shallow (max depth ~100 m), and surface waters are somewhat less saline (~29) than most of the Archipelago due to high river discharge and bounding sills

to the east and west (Williams et al., 2018; Xu et al., 2021). The region is highly nutrient limited, with nitrate concentrations amongst the lowest measured anywhere in the Arctic (Back et al., 2021), however a modest bottom-ice algae bloom still occurs in the spring, peaking in late-May (Campbell et al., 2016).

Our sites were located near the Finlayson Islands, a chain of rocky outcrops about 30 km west of Cambridge Bay (Nunavut, Canada) that bisect Dease Strait in a north-south direction (Fig. 1). These

islands create narrow constrictions where localized currents can be much stronger than is typical for the region. Our study builds on work conducted by Dalman et al. (2019), who described an invisible polynya between the islands. Their study found patterns whereby areas with stronger under-ice currents experienced thinner ice (by approximately 20cm, or 10%) and higher bottom-ice algae biomass (about 2-6 times greater Chl a concentration). Dalman et al. (2019) concluded that stronger current velocities

reduce the ice growth rate by enhancing ocean heat flux to the bottom of the ice and stimulate ice algal growth through turbulent resupply of nutrients. To study the effect of these conditions on the sea ice carbonate system, we selected two level ice sites (Fig. 1) based on Dalman et al. (2019), one approximately 5 km to the east of the islands, outside of the polynya where we expected weaker currents and thick first-year ice (TFYI, within 100 m of their site 5) and one directly between the islands, in the

polynya where we expected strong currents and thin first-year ice (POLY, within 100 m of their site 2).

Sampling was conducted over a 6-day period between 4 and 10 May 2019, with both sites visited on three days (4, 7, and 10 May). On the first sampling date, inverted tilt current meters (Lowell Instruments TCM-1 Current Meter) were installed at both sites, 30 cm below the bottom of the ice. The current meters recorded velocity and temperature for 20 seconds of each minute at a frequency of 16 Hz, which was

averaged to 1-minute intervals in post-processing. The instruments were removed on 10 May, producing a 6-day record of under ice current velocities.

Sea ice sampling was conducted at the sites using a Kovacs Mark II 9-cm diameter coring system. The following general considerations were observed for all cores: snow depth was measured with a meter stick prior to collection; ice thickness was measured after core collection using a Kovacs measuring tape;

when removing cores, the bottoms were pointed away from the sun and shaded to minimize exposure of the ice algal community, and equipment that produced exhaust (snowmobiles, generators) were positioned downwind of the sampling site.

One core was collected at each site on each day for temperature and salinity profiles. Temperature was measured using a Thermopen Mk4 temperature probe inserted into holes drilled at 10-cm intervals. The

core was cut into 10-cm sections, which were sealed in plastic bags and transported back to the field lab where they were melted and analyzed for salinity using a YSI multiparameter probe.

For sea ice carbonate sampling we followed the procedure and materials of Hu et al. (2018). After collecting a core, 10-cm sections were cut in the field, placed into a gas-tight bag, and immediately vacuum sealed using a commercial meat sealer. The sealed samples were transported back to the field lab

and allowed to melt in the dark at room temperature for 15-20 hours. Meltwater was then transferred to 160-mL borosilicate bottles using a peristaltic pump. The pump was operated at low speed, and water was pumped from the bottom of the bag to the bottom of the bottle. The sample bottles were triple rinsed and overflown during filling as per Dickson et al. (2007). Samples were then fixed with a saturated mercuric chloride solution to a final concentration as suggested by Dickson et al. (2007) to prevent





biological activity and sealed using chlorobutyl-isoprene rubber stoppers and aluminium caps (Jiang et al., 2008). Samples were stored in the dark until analysis, which occurred within 3 months of collection.

At each site, we obtained one "high-resolution" core for TIC and TA (10 May), where sections were collected every 10 cm from top to bottom.  The purpose of the high-resolution core was to qualitatively compare the vertical distribution of TIC and TA between the two sites, but as described in Miller et al.

(2015), the inherent heterogeneity of sea ice makes it difficult to quantitatively compare single samples between two sites.  To address this, we designed a statistical sampling method where on each date, 5 locations were randomly selected within a 10-m radius of the centre point of each site. At each of these locations a core was extracted, and we collected the top, bottom, and middle 10 cm for TIC and TA analysis. The middle horizon was determined by measuring the length of each core before sectioning, and

then collecting a section that spanned the centre of the core. The rest of the core volume was discarded. This yielded a total of 15 samples for each horizon (top, middle, bottom) for each site.

To obtain enough sample volume for a bottom ice chlorophyll-a measurement at both sites, the bottom 5 cm of three additional cores were combined ("pooled") in an opaque jug. At each site we collected duplicate samples following this procedure, except on the final sampling day where a single pooled

sample was collected at each site.  Following Dalman et al. (2019), the cores were melted in the cooler jug with filtered seawater added at a 3:1 ratio to reduce osmotic stress on the algae. The melted samples were filtered using 25-mm unburnt GFF filters within 48 hours of collection in a dark room. The filter papers were then packaged in light-impenetrable foil, frozen, and transported to the University of Manitoba for lab analysis.

During our study, we did not collect under-ice seawater samples for carbon system analysis. In past papers, under-ice seawater samples have been used as a reference to compare against sea ice chemistry. Instead, we relied on seawater carbonate system measurements we collected in summer from a small research vessel stationed in Cambridge Bay (see for e.g., Williams et al., 2018). To develop a seawater



reference, we averaged all salinity, TA, and TIC samples collected annually below the summer halocline

from 2016 to 2019 (n=5) at station R3 (68.97°N, 105.47°W) which is within 15 km of the sampling sites. This yielded a seawater reference (or "endmember") of $S_{ref} = 28.8$ (±0.4), $TA_{ref} = 2058$ μmol kg$^{-1}$ (±42), $TIC_{ref} = 1970$ μmol kg$^{-1}$ (±16).

A complimentary survey of snow and ice conditions at the polynya was conducted separately from our primary study. On 12 May, 10 sites at 500-m spacing were visited across the northern end of the polynya,

approximately 1 km north of POLY (Fig. 1). Snow depth was measured at every site, and ice thickness was measured every second site. We use those results in this paper to examine the spatial extent of the invisible polynya.

### 2.3 Laboratory methods

Sea ice melt samples were analyzed at the University of Calgary. TIC was determined through acid

extraction using an automated sample preparation system (AIRICA by Marianda) that quantifies released $CO_2$ using an infrared gas analyzer (LiCOR LI-7000). Four TIC replicates of 1.5 mL were analyzed for each sample; the replicate with the highest coefficient of variation was discarded, and the average of the three remaining replicates was calculated. TA was measured by modified Gran Titration (Grasshoff et al., 1999) using a semi-automated open-cell titration system (AS-ALK2 Apollo SciTech) (Cai et al., 2010).

Replicates were conducted until two measurements with a coefficient of variation of less than 0.002 were obtained and then averaged. Both instruments were calibrated prior to measurement using certified reference material (CRM batch no 177) from the Scripps Institution of Oceanography of the University of California, San Diego. Precision better than ±1.5 μmol kg$^{-1}$ was obtained for TIC, and better than ±3.2 μmol kg$^{-1}$ for TA. Remaining sample water was then analyzed for salinity using an Orion Star A222

conductivity meter paired with a 013010MD conductivity cell, calibrated daily to a NIST traceable

standard.

Chlorophyll *a* concentration was determined by fluorometry. Frozen filters were placed in scintillation

vials with 10 mL of 90% acetone in the dark, briefly agitated on a vortex mixer, and then kept at 4°C for

24 h to extract pigment. Chl *a* fluorescence was then measured (Turner Designs Trilogy fluorometer)

before and after acidifying with 5% HCl following Parsons et al. (1984). From these measurements, the

concentration of Chl *a* was calculated following equations of Holm-Hansen et al. (1965) and a correction

for filtered seawater dilution.

**2.4 Statistical and remote sensing analysis**

Due to the number of samples collected for DIC/TA at the three horizons, and for snow depth and ice

thickness, we were able to test for statistically significant differences between sites using analysis of

variance (ANOVA).  ANOVA tests whether the variance between sample groups (i.e., the two sites) is

greater than the variance within the sample group (i.e., at a single site). In our application, it tests the null

hypothesis that the mean value of any given variable (TIC, TA, snow depth, ice thickness) at the polynya

site (POLY) is equal to the mean at the thick ice site (TFYI).  A p-value is generated, which communicates

the probability of incorrectly rejecting the null hypothesis due to sampling error. ANOVA is a useful

method for relatively small sample sizes (in this case, n~15), because it is robust against modest departures

from normally distributed and homoscedastic sample groups (Burt et al., 2009).

The evolution of sea ice cover during freeze-up was reconstructed from high-resolution (6.25 km

resolution) AMSR2 imagery. Daily sea ice concentration data from October to November 2018 were

obtained from the Universität Bremen open access server (Comiso et al., 2003). The imagery was

georeferenced and followed the threshold of 10% sea ice concentration as the boundary between sea ice

and open ocean (Spreen et al., 2008).



## 3 Results

### 3.1 Physical and biological conditions at the sites

The polynya site, POLY, experienced considerably higher under-ice currents, with peak velocities an order of magnitude higher (20 to 40 cm s$^{-1}$ vs. 2 to 4 cm s$^{-1}$) than outside the polynya, TFYI (Fig. 2 and Table 1). Current velocities at POLY increased throughout the deployment but remained constant at TFYI. Tidal stage predictions (Fig. S1) suggest that the increase at POLY may have been due to a spring tide, but tidal amplitudes are very low in the Cambridge Bay region.

Snow and ice conditions also appeared to be different at the two sites (Table 1) with the mean ice thickness at TFYI about 33 cm (20%) thicker than at POLY. Conversely, the polynya site experienced deeper snow, by an average of 11 cm or about 40%. Ice temperature profiles (Fig. 3) show that on the first two sampling days (4, 7 May) the ice in the polynya was warmer than the thick FYI, by about 2°C at most depths. The temperature at both sites increased on the final sampling day – by 2 to 3°C at the surface and 1 to 1.5°C in the middle of the ice – relative to the first sampling day. This observation is consistent with air temperatures recorded at the Environment Canada weather station in Cambridge Bay, which showed daytime highs increasing from -13°C to -3°C, and nighttime lows increasing from -22°C to -13°C (Fig. S2) during the study period. Ice surface temperatures remained below -6°C on all dates (Fig. 2), and we made no visual observations of melt onset in the snowpack.

In contrast to the results of Dalman et al. (2019), bottom ice algal biomass (reported as Chl *a* concentration) was about 3 times higher at TFYI than at POLY (Table 1). Although we did not collect nearly as many ices algal samples compared to other variables, the range and standard deviations reported in Table 1 show consistency between dates. These results match well with our field observations of much higher visible ice algae at TFYI site on essentially all the cores we extracted.



## 3.2 Carbonate chemistry at the Sites

Results from the high-resolution cores collected on 10 May are shown in Fig. 4. The polynya site was characterized by "c-shaped" profiles for TIC, TA, and salinity, with relatively higher values at the top and bottom of the core compared to the middle. The TFYI site did not have elevated values at the top of the
ice but did display an increase in all parameters in the lowest horizon. The profiles show divergence between sites in the upper and lower ice horizons, but consistency throughout most of the ice volume.

The results from the statistical sampling approach are shown in Table 2 and Table 3, and confirm the general patterns observed in Fig. 4. In the upper ice horizon TIC, TA, and salinity was significantly higher (by 62, 70, and 1.3 $\mu$mol kg$^{-1}$, respectively) at POLY compared to TFYI. The middle horizon was
statistically indistinguishable between the two sites for all three parameters. In the bottom ice horizon, the polynya site experienced significantly lower TIC, TA, and salinity (by 87, 73, and 1.1 $\mu$mol kg$^{-1}$, respectively) than the thick FYI site. TA:TIC ratios are also displayed in Table 2 and 3. These ratios were not statistically different in the top or middle ice horizons but were significantly different in the bottom horizon.


## 4 Discussion

### 4.1 Physical and biological differences between sites

Table 3 shows that the polynya site had statistically thinner ice and thicker snow than the non-polynya site. The strong currents we observed are consistent with a polynya formation mechanism related to
enhanced ocean heat flux, as detailed in Melling et al. (2015). Melling et al. (2015) also hypothesized deep snow as a potential cause of invisible polynyas but noted that variations in snow depth usually occur over scales too large (i.e., hundreds of km) or too small (snow drifts) to be considered polynyas. For our dataset, we cannot rule out snow depth as a contributing factor to the invisible polynya; however, Dalman



et al. (2019) demonstrated ice thickness was significantly different between sites when snow depth was
controlled for during site selection.  Sea ice thermodynamic models (e.g., Flato and Brown, 1996) predict
that maximum ice thickness should vary by about 10% for the difference in snow accumulation that was
observed at the two sites. Such differences are supported by observational studies (e.g., Howell et al.
2016). In our study, the 33cm (20%) thinner ice at the polynya site is probably too great to be explained
by snow depth alone, but it is not unreasonable to estimate that snow may have been responsible for about
half of the difference.

Of course, snow depth is spatially variable over many scales and the differences we observed between
the two sites could have been due to small scale drifting. However, the long transect conducted on 12
May (Fig. 5) shows a consistent pattern of deeper snow and thin ice in the vicinity of the island to the
north (Fig. 1). Local knowledge shared with us by guides is that prevailing winds in this area are from the
northeast.  It is possible that large-scale drifts form on the downwind side of the island (Fig. 1),
contributing to the thinner ice. The timing of ice consolidation is also important, as ice that forms earlier
in the season has longer to accumulate snow. Maps of sea ice concentration that we acquired for the
freeze-up season show that ice at the two sites consolidated at roughly the same time (Fig. 6).

The distribution of ice algal biomass was different from that reported in Dalman et al. (2019) where Chl
$a$ concentrations averaged 9.3 mg m$^{-2}$ at the polynya site and 1.4 mg m$^{-2}$ at the thick ice site (c.f. Table
2).  Not only did we observe higher bottom ice Chl $a$ at TFYI relative to POLY, but we also observed
higher average concentrations there than at any of the sites reported in Dalman et al. (2019). Snow depth
is the dominant limiting factor influencing local ice algal distribution in the Arctic during early spring
(e.g., Welch and Bergmann, 1989; Mundy et al., 2005; Campbell et al., 2015; Lange et al. 2019), with
snow depths > 25 cm limiting ice algal accumulation well into the spring season (Leu et al. 2015).
Therefore, it is most likely that light limitation under thick snow at the polynya site overshadowed any
potential boost in production from current-driven nutrient supply. It is unclear why Dalman et al. (2019)
observed lower Chl $a$ concentrations under a thinner snow cover in a previous year. Photoacclimation





(e.g., Campbell et al., 2015) is one possibility, but with limited data it is difficult to know what the role
of interannual variability is on ice algae biomass.

## 4.2 The upper-ice horizon

Significantly higher salinity in the top 10 cm at the polynya site compared to the thick FYI (Table 3) is
likely a result of initial ice formation conditions.  Typically, higher salinity is associated with faster ice
growth rates (Cox and Weeks, 1974). Sea ice in this region began to form around 15 October and did not
fully consolidate until 7 November. (Fig. 6). Due to the redistribution of drifting ice early in the fall, it is
possible the ice that eventually consolidated in the polynya region did experience a different (potentially
faster) initial growth rate.  However, another explanation is that ice formation at the two sites occurred
under different turbulence conditions, with the polynya site experiencing greater turbulence due to strong
tidal currents.  As reviewed in Weeks (2010), turbulent ice formation conditions promote frazil ice
production, while calm conditions favour congelation growth. In an extensive analysis of sea ice texture
in the Canadian Arctic, Carnat et al. (2013) found the depth of surface frazil to vary between 0 – 15 cm
due to different turbulence regimes. It is possible that the frazil ice layer was thicker at POLY than TFYI,
although confirmation would have required thin section analysis. The randomly oriented nature of frazil
crystals could act to trap more salts and thus yield higher bulk salinity (as we observed at POLY), but this
has not been extensively studied. In a study of Antarctic ice, Eicken (1992) did observe slightly higher
salinities in ice cores characterized by frazil ice, but not high enough to be considered statistically
significant.

The standard deviation of salinity may be a better indication of ice formation conditions; Weeks and Lee
(1962) found the standard deviation of salinity in pancake ice to be greater than 1.0, while congelation
ice had a standard deviation greater than 0.3 – 0.5.  Table 2 shows the standard deviation of salinity we
observed was indeed much higher at the polynya site (1.2) than the thick first year ice site (0.4). Pancake
ice typically forms under significant ocean swell, which we would not expect in these narrow coastal



passages. However, the general concept that turbulent growth conditions lead to higher variability holds for frazil ice (Weeks, 2010) and provides support for our interpretation that salinity variations between
sites in the upper ice horizon was caused by different turbulence regimes. Weeks and Lee (1962) also noted that brine drained much more slowly from pancake ice, which could explain the higher salinity we observed at the polynya site.

At first glance, the statistically higher TIC and TA in surface ice at the polynya site (Table 3) appear to be explainable in the context of bulk salinity and brine rejection. Salinity is a conservative tracer of brine
rejection, and if sea ice carbonate chemistry were only impacted by brine rejection, we would expect bulk TIC and TA to be higher when bulk salinity is higher. Hence the observations between the two sites make sense, and the preceding discussion about processes controlling salinity apply to the carbonate system as well: TIC and TA were probably higher in the surface at the polynya due to ice formation under different turbulence conditions.

However, more nuances become apparent in Fig. 7, which compares the sea ice carbonate system and salinity measurements to a mixing line between a pure ice endmember (salinity, TIC, TA = 0) and our reference seawater endmember (see section 2.2). Figure 7 shows that while the bottom and middle samples collected at both sites follow the mixing line, the top core sections are substantially lower (for both TIC and TA) than the mixing line. The average differences ($\Delta$) between the core samples in the top
horizon and the mixing line were $\Delta$TIC = -118 $\mu$mol kg$^{-1}$, $\Delta$TA = -78 $\mu$mol kg$^{-1}$ at POLY, and $\Delta$TIC = -87 $\mu$mol kg$^{-1}$, $\Delta$TA = -52 $\mu$mol kg$^{-1}$ at TFYI. This indicates a process (or processes) other than brine rejection that affected TA and TIC at the top of the ice.

Calcium carbonate (i.e., ikaite) precipitation lowers both TIC and TA, which can help explain the observed depletions in the surface ice. However, precipitation alone would result in TA being twice as
depleted as TIC, and what we observed is the opposite (TIC was depleted at about twice the rate of TA). Gas evasion is a strong candidate for explaining large depletions in TIC and is consistent with past





measurements of $CO_2$ offgassing by sea ice (e.g., Nomura, Nomura), and observations of TIC loss in the surface layer of bulk ice samples (Miller 2011a). The larger reduction in TIC at the polynya site suggests that ice texture (e.g., frazil vs. congelation ice) may have an impact on $CO_2$ release potential, but to our

knowledge this has not been studied in any detail. It is still possible that ikaite formation occurred in these upper layers and is the most logical explanation for depleted TA relative to the mixing line. For ikaite production to cause a depletion in bulk sea ice TA the crystals must not have been retained in the ice matrix, instead transported to lower ice horizons or the underlying ocean during the rapid brine drainage that occurs in new ice (Vancoppenolle et al. 2006).

**4.3 The middle-ice horizon**

Observations and modeling studies have emphasized a strong relationship between ice growth rate and bulk salinity (e.g., Cox and Week, 1975; Vancoppenolle et al., 2006). Rapidly growing ice tends to trap more brine salt, resulting in higher bulk salinity. We therefore anticipated lower salinity and perhaps lower TA and DIC in the polynya ice where enhanced ocean heat flux (and thicker snow cover) reduced

the ice growth rate. Furthermore, Feltham et al. (2002) proposed that strong under ice currents might enhance brine drainage by inducing pressure variations at the ocean-ice interface, in which case salinity at the polynya site might be even lower. Table 3 shows that our expectations were not met, with no statistical difference in salinity, TA, or TIC observed in the middle horizons. Figure 7 shows that the carbonate species closely followed the conservative mixing line in the middle ice horizon, and that salinity

at both sites fell within a well-defined band of approximately 4 – 5.5. Due to the strength of this salinity relationships, and the similar salinity between sites, we did not observe a significant difference in carbonate system parameters.

Salinity (and hence TA and TIC) may have been similar because the growth rates are not so dramatically different between sites for this portion of the ice volume. Much of the observational and modeling work

relating salinity to ice growth rates has focused on resolving the C-shaped profile of sea ice (e.g., Fig. 4c),





which reflects high brine retention during fast initial ice growth (creating higher salinity in the top horizon than the middle horizon), followed by brine drainage (creating lower salinity in the middle horizons compared to the bottom horizon). Under typical Arctic conditions, the top 20 cm of sea ice can grow in approximately 15 days, for a growth rate of 1.3 cm day$^{-1}$ (e.g., Nakawo and Sinha, 1981). If we assume

sea ice formed in the study area around 15 October (Fig. 6), and the first 20 cm grew by 30 October, that would imply an average "middle horizon" growth rate of about 0.6 cm/day for the polynya site and 0.7 cm/day for the thick FYI site from the end of October to the sampling date. We hypothesize that large difference in initial growth rates compared to later winter growth rates are enough to produce significant vertical variations in salinity, but the small differences in growth rates between the two sites was not

enough to produce noticeable differences in salinity, nor in TIC or TA.

## 4.4 The bottom-ice horizon

Past studies have shown low TIC in the bottom sections of bulk ice melts (Brown et al., 2015) and in brine collected from lower ice horizons (Brown et al., 2015, Delille et al., 2007) in spring sea ice. These lower values are typically attributed to biological drawdown caused by the bottom-ice algal bloom. In our

results, we saw significantly higher TIC and TA (Table 3) at the site with higher Chl a (TFYI, Table 1). This counterintuitive result is explained again by salinity: TFYI had significantly higher salinity than POLY at the bottom (Tables 2, 3), and Figure 7 shows that the bottom ice horizons largely followed the conservative mixing line. It is interesting to note that lower salinity at the site with high current velocity follows the Feltham et al. (2002) prediction that strong under-ice currents drive easier replacement of

high-salinity brine with (relatively) lower salinity seawater. But as noted previously, this is not reflected in the middle ice horizon. Notz and Worster (2009) found that the majority of salt is lost from sea ice through gravity drainage, not at the growing front. Once out of direct contact with the seawater, variables other than current velocity (e.g., thickness, temperature gradient, and brine salinity) may determine salinity (Notz and Worster, 2009). If these factors are relatively consistent between sites, we would expect

salinity (and hence TIC and TA) to also be similar. Nevertheless, it is clear that physical processes (brine





rejection) and not biological processes (i.e., the ice algae bloom) were most responsible for carbonate system differences between the two sites in the bottom layer.

To further check for the impact of primary production on the bottom ice horizon (and any of the other horizons) we normalized TIC and TA values to the estimated seawater endmember salinity ($S_{ref} = 29$)

using $nX = \frac{X_{meas}}{S_{meas}} \times S_{ref}$, where $nX$ is the normalized parameter (TIC or TA), $X_{meas}$ is the bulk ice measurement, and $S_{meas}$ is the measured bulk salinity. This is the appropriate normalization equation for a system where the $S = 0$ endmember is expected to have a TA and TIC equal to 0 µmol kg$^{-1}$ (Friis et al., 2003), which is the case for pure ice. If primary production were significantly affecting any of the ice horizons, we would expect points on a graph of nTA vs. nTIC (Fig. 8) to fall along the

photosynthesis/respiration line. Although statistically significant linear trends ($p < 0.01$) are apparent at both sites, they do not follow the photosynthesis line, even in the bottom ice horizon. In their study of landfast sea ice in the Canadian Arctic Archipelago (Franklin Bay), Miller et al. (2011a) did see a few samples from May that seemed to fall along a photosynthesis line, but most followed a very similar to slope to the ones presented in Fig. 8. Brown et al. (2015) did find substantial TIC drawdown at the ice

bottom due to ice algae blooms in their study near Resolute Bay (also in the Archipelago), but that region experiences bottom ice Chl $a$ value about 10 times higher than either of our sites (Leu et al., 2015). It is possible that the rate of ice algal production in Dease Strait is simply too low to have a noticeable impact on carbonate chemistry.

**4.5 Implications for understanding ikaite in sea ice**

The carbonate chemistry of sea ice has garnered significant attention due to questions surrounding ikaite precipitation and dissolution. Our preceding analysis of the three ice horizons only invoked ikaite





precipitation to explain patterns observed in the top horizon, where it seemed a secondary process to gas exchange. So, is ikaite precipitation apparent (or important) in our samples?

The primary evidence presented by Rysgaard et al. (2007) and Rysgaard et al. (2009) to support the hypothesis of ikaite precipitation was the observation of TA:TIC ratios greater than 1:1 and approaching 2:1 in some samples. The only relatively high TA:TIC ratios that we observed were in the upper ice horizon (Table 2), which other studies (Miller et al. 2011a; Geilfus et al. 2013) have observed and attributed primarily to outgassing from the ice surface. But average TA:TIC ratios in our samples were close to 1.05 in the bottom and middle ice horizons (Table 1), and not meaningfully different from our estimate of the source seawater TA:TIC ratio (1.04). While the original formulation of the sea ice pump hypothesis (Rysgaard et al., 2007) reported bulk TA:TIC ratios approaching 2:1, the majority of studies since have reported values close to 1:1 below the surface ice layers (Miller et al., 2011a, 2011b; Geilfus et al., 2012; Brown et al., 2015; Findlay et al., 2015).

Of course, the TA:TIC ratio is not the only available evidence of ikaite precipitation in sea ice. There is microscopic evidence (e.g., Rysgaard et al., 2013), and crystals extracted from sea ice have been positively identified by x-ray diffraction (Dieckmann, et al. 2008). We did not have access to such techniques, but Figure 8 provides evidence that ikaite precipitation did occur in our study. At both sites, a linear relationship between nTIC and nTA with a slope close to 1:1 (0.8 at TFYI, 0.7 at POLY), was observed through all samples. These slopes are nearly identical to those observed by Miller et al. (2011a), who hypothesized that such a relationship is the result of combined ikaite precipitation and respiration in the ice. Similar slope values were observed by Geilfus et al. (2012) and Brown et al. (2015) in brine. Therefore, ikaite precipitation certainly seems to be occurring in our study area; our contention is simply that this location (and apparently at several others) the amount of ikaite precipitation is not enough to result in a large fractionation of TA and TIC within the sea ice.



The well-known spatial heterogeneity of sea ice has often been invoked to explain differences between studies, including differences in the TA:TIC ratio (compare for example, Rysgaard et al., 2007 and Miller et al., 2011a). Due to the scale of brine channels and the size of the standard core barrel, it is not uncommon for two ice cores sampled essentially side-by-side to produce remarkably different results for any biogeochemical variable (Miller et al., 2015). To our knowledge, the present study is the first attempt
to employ a statistical sampling approach to compare sea ice carbonate chemistry between two sites. What our results show is that despite large differences in the physical and biological characteristics between two locations, when small-scale heterogeneity is accounted for by averaging the results can be surprisingly similar. Table 4 shows that when a weighted vertical average of the two sites is calculated, the carbonate chemistry (particularly the TA:TIC ratio) is not meaningfully different; the statistically
significant differences in the thin top and bottom layers are outweighed by the thick middle layer. Our study, designed to examine extremes in spatial variability, suggest that we should perhaps be looking at other factors (i.e., methodologies) to explain divergent reports of sea ice carbonate chemistry.

Setting aside the TA:TIC ratio, the actual values of bulk ice TIC and TA (Table 4) that we observed are fairly consistent with other measurements from the Canadian Arctic Archipelago. Rysgaard et al. (2009),
Miller et al. (2011a), Findlay et al. (2015) and Brown et al. (2015) all report values between ~300-400 μmol kg$^{-1}$ for TA, and 350-450 μmol kg$^{-1}$ for TIC. Higher values (up to ~700 μmol kg$^{-1}$ for both TIC and TA) have been reported in deep Arctic basins and around Greenland (Rysgaard et al., 2009). Salinity, and hence carbonate species concentrations, tend to be lower in the Canadian Arctic Archipelago due to the large volume of river runoff that impacts the region. It is therefore likely that the source water from which
sea ice grows plays a significant role in winter sea ice carbonate concentrations.

## 5 Conclusions

We contrasted the bulk sea ice carbonate chemistry of an invisible polynya site (strong currents, thin ice,



thick snow, lower algal chl-*a*) with a nearby site outside of the polynya (weak currents, thick ice, thin
snow, higher algal chl-*a*). Statistically significant differences in chemistry were noted between the two
locations, but only in thin surface and bottom layers. The middle ice horizons – which represent about
85% of the ice volume – had statistically indistinguishable salinity, TIC, and TA concentrations. The
observation that two sites with such different physical and biological characteristics could yield such
similar carbonate system chemistries suggests that processes known to modify the sea ice carbonate
system are either less impactful, or less variable, than expected. While our results do show evidence of
some combination of gas evasion, respiration, and ikaite precipitation impacting chemistry, these
processes were dominated by brine drainage processes as evidenced by strong relationships with salinity.
Although considerable regional variability in bulk ice TIC and TA concentrations are evident when
comparing our results to other studies, strong salinity relationships are a consistent result. Therefore, it
may be possible to derive an empirical relationship between sea ice salinity and the source seawater
chemistry to predict sea ice carbonate chemistry; an outcome which may have utility in modeling studies.
It is important to note, however, that the geographical areas currently represented in the literature are
fairly limited in scope. We expect regional variability to play an important role; for example the
differences we measured in ice algae biomass are considerably lower that the difference between our sites
and somewhere like Resolute Bay. An ongoing effort by the Biogeochemical Exchange Processes at Sea
Ice Interfaces (BEPSII) working group to compile published and unpublished data may reveal
complexities that we cannot capture in a single Arctic region. Such an effort may also shed light on the
important (and unresolved) question of how best to characterize the TA:TIC ratio in sea ice.



**Acknowledgements**

Thanks to Shawn Marriott, Zoe Walker, and the staff of the Canadian High Arctic Research Station who assisted with fieldwork. Alexandre Langois provided the long snow depth/sea ice thickness transect data. We acknowledge the ongoing logistical support of Polar Knowledge Canada. Financial support was provided by the Marine Environmental Observation Prediction and Response (MEOPAR) Network of 515 Centres of Excellence, the ArcticNet Network of Centres of Excellence, the Canada Foundation for Innovation John R. Evans Leaders Fund, and the University of Calgary. A. Cranch was supported by a scholarship from the Northern Science Training Program (NSTP). This paper is a contribution to the SCOR Working Group 152 – Measuring Essential Climate Variables in Sea Ice (ECV-Ice).




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





**Table 1:** Mean ($\bar{x}$), standard deviation ($\sigma$) and range (max-min) of the measured physical and biological conditions at the two sampling sites. Sample size for snow depth and ice thickness was *n*=15, except for TFYI ice thickness (*n*=14). Sample size for Chl *a* was *n*=3, therefore standard deviation is not displayed.

| | Thick FYI (TFYI) | | | | Thin FYI (POLY) | | | |
|---|---|---|---|---|---|---|---|---|
| | Snow Depth (cm) | Ice Thickness (cm) | Current Velocity (cm/s) | Bottom Chl *a* (mg/m$^2$) | Snow Depth (cm) | Ice Thickness (cm) | Current Velocity (cm/s) | Bottom Chl *a* (mg/m$^2$) |
| $\bar{x}$ | 15.0 | 180.5 | 1.4 | 14.2 | 26.0 | 146.8 | 9.5 | 4.3 |
| $\sigma$ | 4.8 | 8.9 | 0.6 | | 7.9 | 16.2 | 9.5 | |
| range | 8.0 - 25.0 | 161 - 191 | 0.5 - 4.3 | 11.3 - 16.1 | 9.5 - 26.0 | 124 - 174 | 0.0 - 44.8 | 2.6 - 5.6 |

**Table 2:** Summary of biogeochemical variables measured at the two sites, across the three horizons. Mean ($\bar{x}$) and standard deviation ($\sigma$) are presented. Sample depth is the mid-point of the core sections collected for each horizon. For each horizon, at each site, *n*=15.

| | Thick FYI (TFYI) | | | | Thin FYI (POLY) | | | |
|---|---|---|---|---|---|---|---|---|
| | Snow Depth (cm) | Ice Thickness (cm) | Current Velocity (cm/s) | Bottom Chl *a* (mg/m$^2$) | Snow Depth (cm) | Ice Thickness (cm) | Current Velocity (cm/s) | Bottom Chl *a* (mg/m$^2$) |
| $\bar{x}$ | 15.0 | 180.5 | 1.4 | 14.2 | 26.0 | 146.8 | 9.5 | 4.3 |
| $\sigma$ | 4.8 | 8.9 | 0.6 | | 7.9 | 16.2 | 9.5 | |
| range | 8.0 - 25.0 | 161 - 191 | 0.5 - 4.3 | 11.3 - 16.1 | 9.5 - 26.0 | 124 - 174 | 0.0 - 44.8 | 2.6 - 5.6 |






**Table 3:** Results of the ANOVA analysis for parameters measured at the two sites. Shaded cells indicate a statistically significant ($p < 0.01$) difference in the mean value between the two sites.

| | | Variable Tested | ANOVA p-value |
|---|---|---|---|
| | | Snow Depth | <0.001 |
| | | Ice Thickness | <0.001 |
| HORIZON | Top | Salinity | <0.001 |
| | | TIC | 0.001 |
| | | TA | 0.003 |
| | | TA:TIC | 0.544 |
| | Middle | Salinity | 0.339 |
| | | TIC | 0.388 |
| | | TA | 0.218 |
| | | TA:TIC | 0.252 |
| | Bottom | Salinity | <0.001 |
| | | TIC | <0.001 |
| | | TA | <0.001 |
| | | TA:TIC | 0.001 |

**Table 4:** Vertically weighted average TIC and TA for the two sites. Weighting calculated attributing top
and bottom horizon values (Table 2) to the surface and bottom 10cm of ice, and the middle horizon values
to the mean ice thickness subtracting the 20 cm accounted for by the top and bottom measurements.

| | $\overline{\text{TIC}}$ ($\mu$mol kg$^{-1}$) | $\overline{\text{TA}}$ ($\mu$mol kg$^{-1}$) | TA:TIC |
|---|---|---|---|
| TFYI | 340.24 | 356.60 | 1.05 |
| POLY | 349.42 | 373.84 | 1.07 |
| Difference | 2.6% | 4.6% | 2.0% |

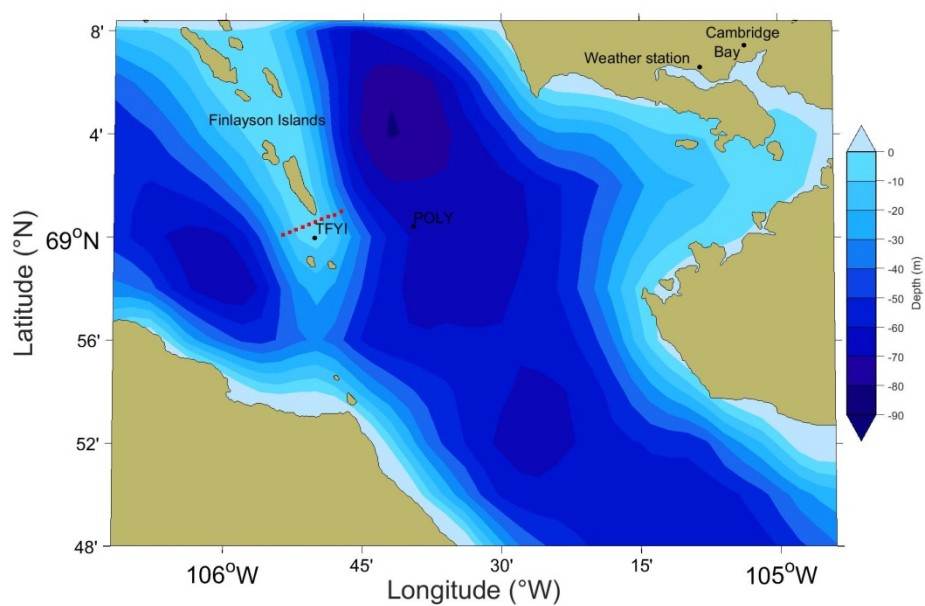


**Figure 1: Study area (including bathymetry), showing the location of the two sampling sites. The thick first year ice site (TFYI) was located at 69.01°N, 105.66°W. The polynya site (POLY) was located at 68.99°N, 105.84°W. The proximity to Cambridge Bay, the nearest Environment Canada**
**weather station, and the snow depth/ice thickness transect (red squares) is also shown.**





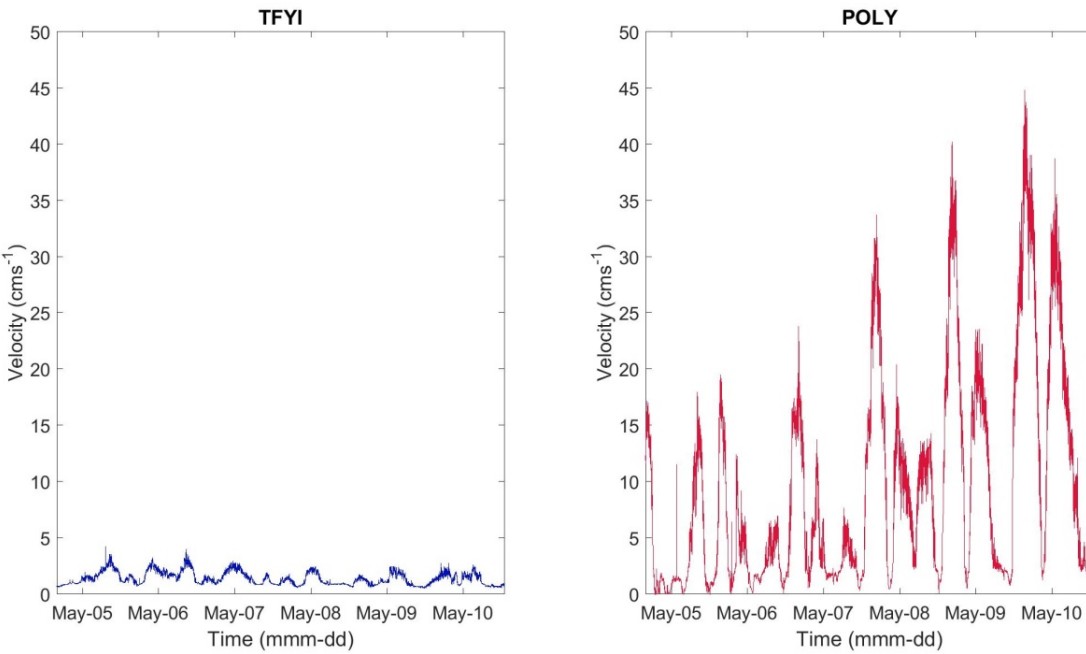

**Figure 2: Under ice current velocities observed at the two sites during the sampling period.**





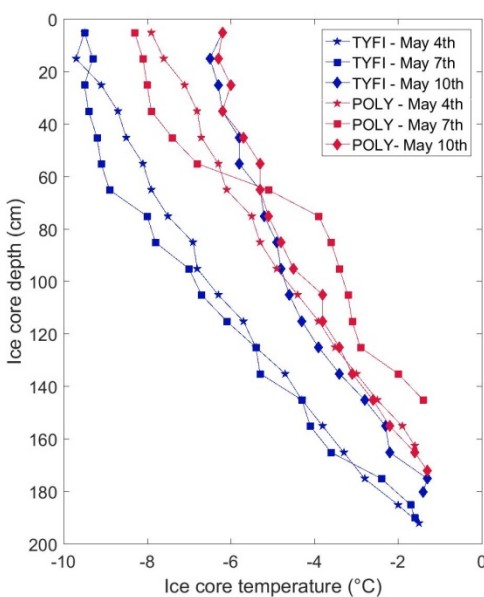


**Figure 3: Ice temperature profiles at both sites, from each of the daily temperature/salinity cores.**





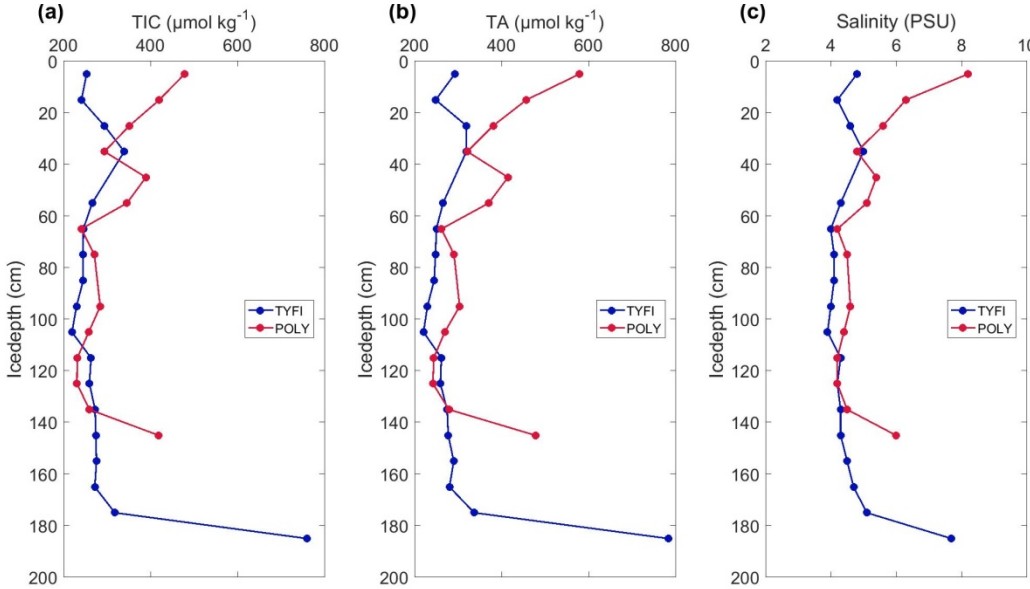

**Figure 4: Results of the high resolution (10-cm interval) cores collected on the final sampling day.**






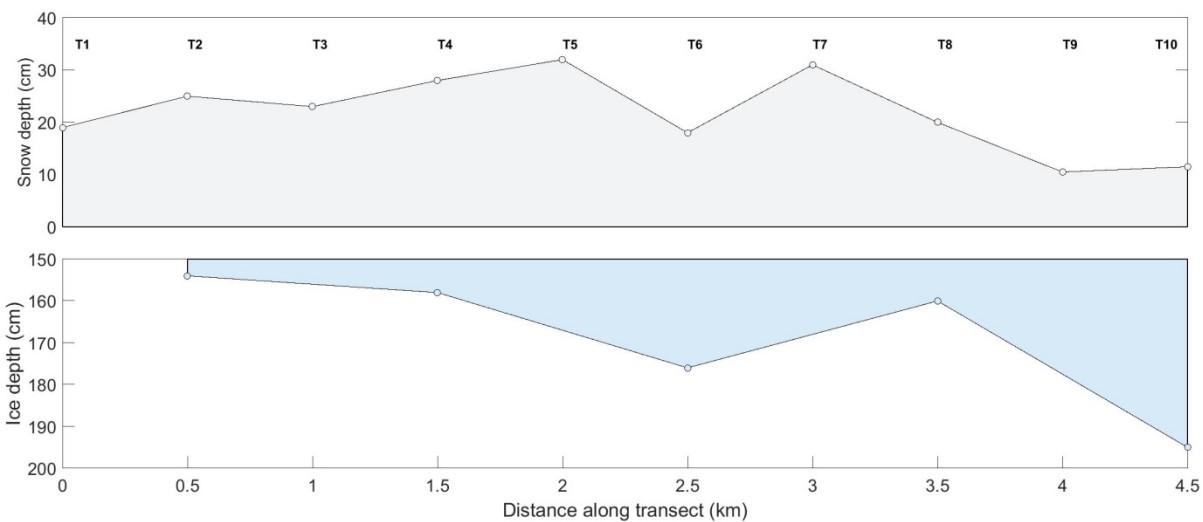

**Figure 5: Results of snow depth and ice thickness survey conducted just north of our sampling sites on 12 May.**




**Figure 6: Sea ice concentration of the study area during sea ice freeze-up on (a) 19 October, (b) 27 October, (c) 2 November, and (d) 7 November 2018. Sites of interest within this study are denoted as black and white dots.**





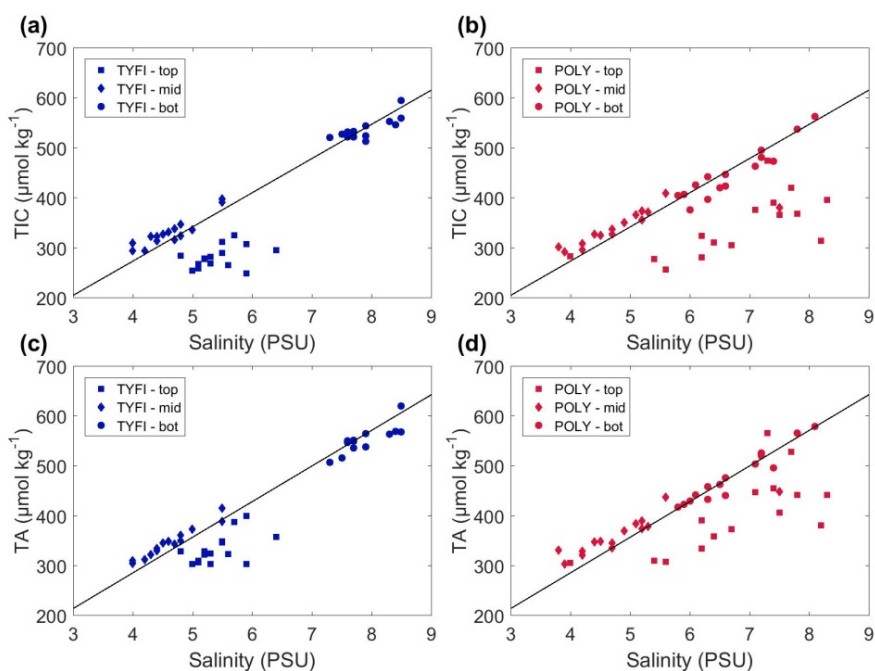

**Figure 7: Bulk Sea ice TIC and TA plotted against bulk salinity for both sites.**



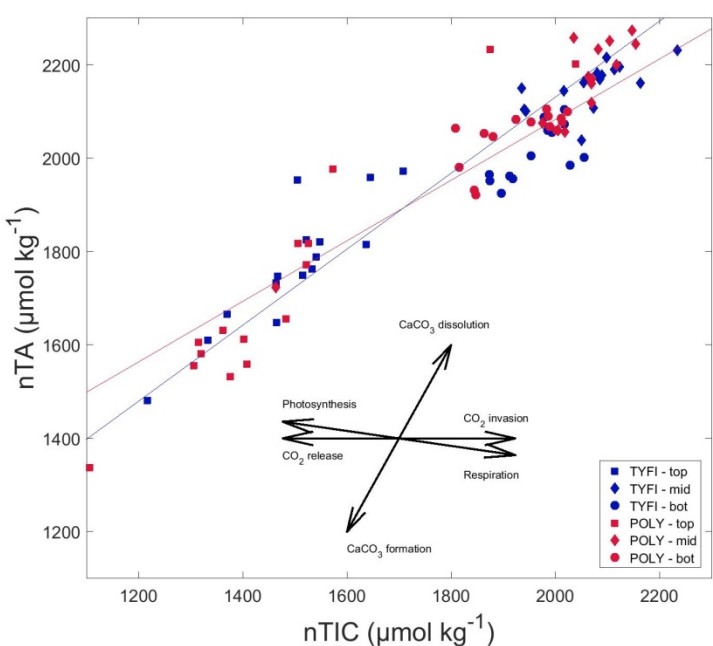

**Figure 8: Normalized bulk TIC (nTIC) plotted against normalized bulk TA (nTA).**