# Peer review of "Variability in sea ice carbonate chemistry: A case study comparing the importance of ikaite precipitation, bottom ice algae, and currents across an invisible polynya"

_The Cryosphere, 2021_

## Referee Comment (RC1)

Review of the paper entitled "*Variability in sea ice carbonate chemistry: A case study comparing the importance of ikaite precipitation, bottom ice algae, and currents across an invisible polynya*" by Else and co-authors

**General comments:**

I commend the authors on a very interesting paper describing the variability in the sea ice carbonate chemistry between 2 Arctic sites, one of which being an invisible polynya. In particular the authors collected several cores at each site to perform a statistical comparison of the carbonate chemistry in the various sea ice horizons. They observed differences in the upper and bottom part of the sea ice mostly. The authors also observed that biogeochemical processes such as air-ice $CO_2$ fluxes, ikaite precipitation and primary production only played a little role in the distribution of carbonate species, and that physical processes linked to ice growth and brine drainage played the most important role. This is a very interesting and nicely written paper that contributes to the understanding of carbonate chemistry in the sea ice and, thus, the role of sea ice in the carbon cycle of polar regions.

I give below numerous minor comments to improve the manuscript. Therefore, I suggest publication in The Cryosphere if the authors can improve the few minor corrections given below. I hope these comments will help the authors to strengthen this very interesting manuscript, as it will bring a significant contribution to the understanding of the sea ice biogeochemistry and the role of sea ice in the carbon cycle of polar regions.

Best regards,

Sébastien Moreau
Norwegian Polar Institute
Tromsø, 9007, Norway

**Specific comments:**
**Abstract:**

Line 31: "*biology did not have a noticeable impact*", in fact as I commented below in the discussion, it is possible that primary production and respiration had opposite effects on the carbonate chemistry at the bottom of the sea ice, which led to this lack of detectable effects of biology on the carbonate chemistry. I suggested below that you investigate regressions lines only for the bottom ice section in your Figure 8.

**Introduction**
The introduction is very nicely organized, nicely written and a pleasure to read.

Line 89: we did show that this outflux of CO2 at the surface of the ice was due to the lack of permeability of sea ice below this few centimeter-thick layer (Moreau et al., 2015). Moreau, S., M. Vancoppenolle, B. Delille, J.-L. Tison, J. Zhou, M. Kotovitch, D. N. Thomas, N.-X. Geilfus, and H. Goosse (2015), Drivers of inorganic carbon dynamics in first-year sea ice: A model study, J. Geophys. Res. Oceans, 120, doi:10.1002/2014JC010388.

Line 108: needs a space "*31in the Arctic*"

Line 115: "*in all cases, this limited effect was primarily due to the lack of deep brine convection in winter most Arctic and Antarctic regions*". This sentence is a bit misleading I find. For the Southern Ocean, oceanographic studies have only identified the formation of Antarctic Bottom waters (AABW) in four localized areas which are the Ross Sea, the Metz polynya, Cape Darnley and the southwestern Weddell Sea. Both NEMO-LIM and MPIOM/HAMOCC produce deep winter convection events in these areas in the Arctic and Southern Oceans, which is consistent with the following sentence (line 115 to 118). So perhaps you could rephrase this sentence a bit.

**Materials and Methods**

Line 145: can you give a range of the nitrate concentrations typically observed there?

Line 184: what is the final HgCl2 concentration?

**Results**

In Table 1, the bottom Chl-a is reported in mg/m2 instead of concentration. Can you explain in the Materials and Methods section how you converted the measured Chl-a concentration to an integrated biomass?

Table 2 is the same as Table 1 while I understand it should present the summary of biogeochemical variables.

**Discussion**

Line 317: here you refer to Chl-a concentrations rather than biomass, so I would suggest to keep it consistent, using either concentrations or biomass throughout the manuscript.

Line 327-342: could surface flooding of sea ice be another explanation for the higher salinity, TIC and TA observed in the POLY site? Or do you think it is not a plausible explanation?

Line 372: miss the citations dates in "*(e.g., Nomura, Nomura)*"

Line 360-379 and Figure 7: this is a very interesting result and challenges our typical view of the precipitation of ikaite crystals in sea ice. I agree with you that ikaite crystals must have been displaced to explain the lower TA with respect to salinity in the upper part of the ice. Perhaps you should also explain to readers that, if present, ikaite crystals would have dissolved during the melting of the ice, which would have taken the measured TA values closer to the theoretical dilution line.

Line 433: remove to from "*a very similar to slope to the ones*"

Line 436-438: it's also possible that bacterial respiration acted in the opposite way than photosynthesis, keeping the TIC values higher. Perhaps you could also mention this hypothesis.

In fact, when looking at Figure 8, I wonder if the regression lines would be different if only considering the bottom ice sections? Would it be closer to the theoretical line for photosynthesis/respiration effects on TIC and TA? Perhaps you could add these specific bottom sea ice regression lines to the Figure as well?

Line 458 and Figure 8: could you also indicate the values of the slopes for the regression lines on the figure itself?

Line 464: this is then probably due to the loss of the ikaite crystals which you described convincingly in the first section of the discussion.

Line 472-473: this sentence is missing words it seems "*when small-scale heterogeneity is accounted for by averaging the results can be surprisingly similar.*"

**Figures:**

Figure 1: a zoom-out insert would be nice to have to place Cambridge Bay on a larger map. In addition, the text is very small in the figure, and so are the transect and stations dots.

---

## Author Comment (AC1)

**Review RC1 (Sébastien Moreau)**

Review of the paper entitled "Variability in sea ice carbonate chemistry: A case study comparing the importance of ikaite precipitation, bottom ice algae, and currents across an invisible polynya" by Else and co-authors

**General comments:**

I commend the authors on a very interesting paper describing the variability in the sea ice carbonate chemistry between 2 Arctic sites, one of which being an invisible polynya. In particular the authors collected several cores at each site to perform a statistical comparison of the carbonate chemistry in the various sea ice horizons. They observed differences in the upper and bottom part of the sea ice mostly. The authors also observed that biogeochemical processes such as air-ice $CO_2$ fluxes, ikaite precipitation and primary production only played a little role in the distribution of carbonate species, and that physical processes linked to ice growth and brine drainage played the most important role. This is a very interesting and nicely written paper that contributes to the understanding of carbonate chemistry in the sea ice and, thus, the role of sea ice in the carbon cycle of polar regions. I give below numerous minor comments to improve the manuscript. Therefore, I suggest publication in The Cryosphere if the authors can improve the few minor corrections given below. I hope these comments will help the authors to strengthen this very interesting manuscript, as it will bring a significant contribution to the understanding of the sea ice biogeochemistry and the role of sea ice in the carbon cycle of polar regions.

Best regards,
Sébastien Moreau
Norwegian Polar Institute
Tromsø, 9007, Norway

**Specific Comments:**

Abstract:
**RC1.01** Line 31: "biology did not have a noticeable impact", in fact as I commented below in the discussion, it is possible that primary production and respiration had opposite effects on the carbonate chemistry at the bottom of the sea ice, which led to this lack of detectable effects of biology on the carbonate chemistry. I suggested below that you investigate regressions lines only for the bottom ice section in your Figure 8.

*We took this suggestion, and it significantly aided our interpretation of the bottom ice horizon. Please see our response to comment RC1.14, but more importantly our response to Reviewer 2's comment RC2.01, which was similar. For the abstract, we adjusted the line in question to read:*

*"Carbonate chemistry in the bottom horizon could largely be explained by the salinity distribution, with the strong currents at the polynya site potentially playing a role in desalinisation; biology appeared to exert only a minor control, with some evidence that the ice algae community was net heterotrophic."*

Introduction
The introduction is very nicely organized, nicely written and a pleasure to read.

**RC1.02** Line 89: we did show that this outflux of CO2 at the surface of the ice was due to the lack of permeability of sea ice below this few centimeter-thick layer (Moreau et al., 2015). Moreau, S., M. Vancoppenolle, B. Delille, J.-L. Tison, J. Zhou, M. Kotovitch, D. N. Thomas, N.-X. Geilfus, and H.

Goosse (2015), Drivers of inorganic carbon dynamics in first-year sea ice: A model study, J. Geophys. Res. Oceans, 120, doi:10.1002/2014JC010388.

*Thanks, this paper does present a good discussion of the controls of outgassing. Manuscript has been revised to read:*

*"During ice formation, some brine is rejected upwards, allowing $CO_2$ to offgass to the atmosphere (Geilfus et al., 2013; Nomura et al., 2014). In the frazil crystal structures of surface ice, the high brine fraction creates high enough permeability (relative to layers immediately below it) for this outgassing to occur over a layer several centimetres thick (Moreau et al., 2015)."*

**RC1.03** Line 108: needs a space "31in the Arctic"

*Correction made.*

**RC1.04** Line 115: "in all cases, this limited effect was primarily due to the lack of deep brine convection in winter most Arctic and Antarctic regions". This sentence is a bit misleading I find. For the Southern Ocean, oceanographic studies have only identified the formation of Antarctic Bottom waters (AABW) in four localized areas which are the Ross Sea, the Metz polynya, Cape Darnley and the southwestern Weddell Sea. Both NEMO-LIM and MPIOM/HAMOCC produce deep winter convection events in these areas in the Arctic and Southern Oceans, which is consistent with the following sentence (line 115 to 118). So perhaps you could rephrase this sentence a bit.

*Yes, we see the point the reviewer is making. That sentence did leave the wrong impression. We have revised this section to read:*

*"Using a one-dimensional model with more detailed biogeochemical processes, Mortenson et al. (2018) similarly found that ikaite precipitation and dissolution did not have a large impact on annual air-sea $CO_2$ flux budgets due to a lack of deep convection at their study site. However, both Grimm et al. (2016) and Moreau et al. (2016) concluded that the sea ice carbon pump is important in specific polar regions (i.e., deep water formation areas), and could be important on long timescales (e.g., during the glacial/interglacial transitions)."*

Materials and Methods
**RC1.05** Line 145: can you give a range of the nitrate concentrations typically observed there?

*Yes, we have added the following:*

*"The region is severely nutrient limited, with nitrate concentrations amongst the lowest measured anywhere in the Arctic (1.0 – 1.2 mmol $L^{-1}$ beneath the pycnocline in summer (Back et al., 2021); 1.3 mmol $L^{-1}$ beneath sea ice (Dalman et al., 2019)), however a modest bottom-ice algae bloom still occurs in the spring, peaking in late-May (Campbell et al., 2016)."*

**RC1.06** Line 184: what is the final HgCl2 concentration?

*We added the following, which is consistent with how the Dickson SOP describes HgCl additions:*

*"Samples were then fixed with 80 uL of saturated mercuric chloride (0.05% of the total sample volume, as per Dickson et al., (2007)) to prevent biological activity, and sealed using chlorobutyl-isoprene rubber stoppers and aluminium caps (Jiang et al., 2008)."*

Results
**RC1.07** In Table 1, the bottom Chl-a is reported in mg/m2 instead of concentration. Can you explain in the Materials and Methods section how you converted the measured Chl-a concentration to an integrated biomass?

*We have added the following note to the methods section:*

*"Concentrations were converted to biomass (e.g. Chl a expressed as mg $m^{-2}$) by multiplying by the length of the core section collected (in this case, 5 cm or 0.05 m)."*

**RC1.08** Table 2 is the same as Table 1 while I understand it should present the summary of biogeochemical variables.

*Wow, we are very sorry about that. The updated table should be as follows:*

**Table 2:** Summary of biogeochemical variables measured at the two sites, across the three horizons. Mean ($\bar{x}$) and standard deviation ($\sigma$) are presented. Sample depth is the mid-point of the core sections collected for each horizon. For each horizon, at each site, *n*=15.

| HORIZON | | | Thick FYI (TFYI) | | | | | Polynya (POLY) | | | | |
|---|---|---|---|---|---|---|---|---|---|---|---|---|
| | | | Sample Depth (cm) | Salinity (PSU) | TIC ($\mu$mol kg$^{-1}$) | TA ($\mu$mol kg$^{-1}$) | TA:TIC | Sample Depth (cm) | Salinity (PSU) | TIC ($\mu$mol kg$^{-1}$) | TA ($\mu$mol kg$^{-1}$) | TA:TIC |
| | Top | $\bar{x}$ | 5 | 5.43 | 281.63 | 333.24 | 1.18 | 5 | 6.79 | 343.33 | 403.56 | 1.17 |
| | | $\sigma$ | 0 | 0.42 | 21.98 | 30.16 | 0.05 | 0 | 1.18 | 61.83 | 78.13 | 0.05 |
| | Middle | $\bar{x}$ | 89 | 4.63 | 331.55 | 345.91 | 1.05 | 73 | 4.88 | 341.89 | 363.22 | 1.06 |
| | | $\sigma$ | 4 | 0.46 | 29.77 | 31.39 | 0.03 | 8 | 0.90 | 34.60 | 40.88 | 0.04 |
| | Bottom | $\bar{x}$ | 176 | 7.89 | 538.37 | 551.60 | 1.02 | 142 | 6.73 | 450.93 | 478.71 | 1.06 |
| | | $\sigma$ | 9 | 0.39 | 20.55 | 26.08 | 0.02 | 16 | 0.71 | 52.45 | 51.65 | 0.03 |

Discussion
**RC1.09** Line 317: here you refer to Chl-a concentrations rather than biomass, so I would suggest to keep it consistent, using either concentrations or biomass throughout the manuscript.

*Thank you, good point. Given that we are reporting in mg m-2, we have made the adjustment to biomass throughout the document.*

**RC1.10** Line 327-342: could surface flooding of sea ice be another explanation for the higher salinity, TIC and TA observed in the POLY site? Or do you think it is not a plausible explanation?

*It's plausible enough to be worth mentioning. We have added the following:*

*"Another potential cause of differences at the surface between sites is flooding, the process that occurs when relatively thick snow depresses the ice surface below the freeboard level, allowing seawater to flow over the ice. Flooding is less common in the Arctic than the Antarctic due to relatively thinner snowpacks and thicker ice (Provost et al., 2017), and during our study period we observed positive freeboard at all sites. But given the deeper snowpack at POLY, it is possible that flooding occurred earlier in the season when the ice was still quite thin. As described by Eicken et al. (1992), this would lead to higher surface salinity."*

**RC1.11** Line 372: miss the citations dates in "(e.g., Nomura, Nomura)"

*This has been corrected, and the following references have been added to the reference list:*

*Nomura, D., Yshikawa-Inoue, H., Toyota, T.: The effect of sea ice growth on air-sea CO2 flux in a tank experiment, Tellus B: Chemical and Physical Meteorology, 58(5) 418-426, 2006.*
*Nomura, D., Yoshikawa-Inoue, H., Toyota, T., Shirasawa, K.: Effects of snow, snowmelting and refreezing processes on air-sea ice CO$_2$ flux, Journal of Glaciology, 56(196) 262-270, 2010.*

**RC1.12** Line 360-379 and Figure 7: this is a very interesting result and challenges our typical view of the precipitation of ikaite crystals in sea ice. I agree with you that ikaite crystals must have been displaced to explain the lower TA with respect to salinity in the upper part of the ice. Perhaps you should also explain to readers that, if present, ikaite crystals would have dissolved during the melting of the ice, which would have taken the measured TA values closer to the theoretical dilution line.

*We thought it would be best to add this to the methods section, and added the following text:*

*"At room temperature, ikaite crystals are highly unstable and dissolve back into the meltwater, which is why samples collected using this method are best described as characterizing "total" (as opposed to dissolved) inorganic carbon."*

**RC1.13** Line 433: remove to from "a very similar to slope to the ones"

*Correction made.*

**RC1.14** Line 436-438: it's also possible that bacterial respiration acted in the opposite way than photosynthesis, keeping the TIC values higher. Perhaps you could also mention this hypothesis.
In fact, when looking at Figure 8, I wonder if the regression lines would be different if only considering the bottom ice sections? Would it be closer to the theoretical line for photosynthesis/respiration effects on TIC and TA? Perhaps you could add these specific bottom sea ice regression lines to the Figure as well?

*Reviewer 2 asked made a similar suggestion, and it is a very good point. I will summarize our response here, but please see our response to comment RC2.01 for complete details:*

*-It was difficult to put all the regression lines on Fig. 8, so we made a new table (now Table 4) which reports the slope at each horizon. You can find this table in our response to RC2.01.*
*-In sections 4.4 and 4.5, we included a discussion of the different ice horizons, which did clarify the role of biology (actually suggesting a stronger impact of respiration) on the bottom ice horizon. The new text is in our response to RC2.01.*

*This was a very useful suggestion, we are glad both reviewers brought it up.*

**RC1.15** Line 458 and Figure 8: could you also indicate the values of the slopes for the regression lines on the figure itself?

*Correction made*

**RC1.16** Line 464: this is then probably due to the loss of the ikaite crystals which you described convincingly in the first section of the discussion.

*This section has been changes slightly in response to Reviewer 2's comment RC2.01. However, we have also added the following sentence which directly addresses this point:*

*"Or perhaps more precisely, ikaite crystals are lost to the underlying seawater at a similar rate to brine rejection."*

**RC1.17** Line 472-473: this sentence is missing words it seems "when small-scale heterogeneity is accounted for by averaging the results can be surprisingly similar."

*We have re-written this sentence as:*

*"What our results show is that despite large differences in the physical and biological characteristics between two locations, when small-scale heterogeneity is accounted for by averaging the results can be surprisingly similar."*

Figures:
**RC1.18** Figure 1: a zoom-out insert would be nice to have to place Cambridge Bay on a larger map. In addition, the text is very small in the figure, and so are the transect and stations dots.

*This figure and its caption have been revised to address these suggestions:*

[Figure]

**Figure 1: Study area (including bathymetry), showing the location of the two sampling sites. The thick first year ice site (TFYI) was located at 69.01°N, 105.66°W. The polynya site (POLY) was located at 68.99°N, 105.84°W. The proximity to Cambridge Bay (white star), the nearest Environment Canada weather station (white diamond), and the snow depth/ice thickness transect (red circles) is also shown.**

---

## Author Comment (AC2)

**Review RC2 (Eric Mortenson)**

General comments
The study presents a comparison of the ice carbonate system throughout the ice column between two sites exhibiting different biological (ice algal) and physical (currents, snow cover) conditions. One of the main results is emphasis on the importance of brine drainage over secondary processes like primary productivity and ikaite precipitation in characterizing the ice-carbonate system. They acknowledge that air-sea exchange and ikaite precipitation may be important near the surface, and that primary productivity be important near the bottom, but the much thicker middle of the ice column has a stronger influence when averaging over the entire ice column.

**RC2.01** The challenge to the main result above is the TIC and TA deficits in the surface ice, as shown in the S:TIC and S:TA plots (fig 7). Although this layer is small relative to the entire ice column (therefore making the deficits not as important when considering the entire ice column), why are the nDIC:nTA slopes (fig 8) so close to the ikaite precipitation slope?

I think the answer is that the different horizons are improperly weighted in fig 8. I.e., with the exception of the "high-resolution" core, there are an equal number of measurements for the relatively thin surface and bottom layers as there are for the thicker middle layer. I suggest making separate regression lines for each layer for the 2 cores, and stating that the thicker middle layer dominates the bulk characteristics (per unit area).

*This is a very good point, and one we hadn't considered when constructing and interpreting Fig. 8. In response to your comment we attempted to draw regression lines on Fig. 8 for each of the horizons, but it became cluttered. Instead, we created a new table (Table 4) which reports the slopes for each horizon for each of the sites, along with some diagnostic statistics:*

**Table 4:** Slope values of nTA/nTIC (see Figure 8) for each of the horizons at each site, and for all samples at each site. Also reported is the $r^2$ value for each regression, as well as a p-value testing the significance of each slope (against the null hypothesis that slope = 0).

| | | Thick FYI (TFYI) | | | Polynya (POLY) | | |
|---|---|---|---|---|---|---|---|
| | | Slope | $r^2$ | p value | Slope | $r^2$ | p value |
| HORIZON | Top | 0.96 | 0.79 | <0.01 | 1.03 | 0.91 | <0.01 |
| | Middle | 0.37 | 0.39 | 0.01 | 0.87 | 0.86 | <0.01 |
| | Bottom | 0.67 | 0.46 | <0.01 | 0.50 | 0.48 | <0.01 |
| | All | 0.65 | 0.88 | <0.01 | 0.81 | 0.95 | <0.01 |

*To discuss this new table, and to address the point about the middle layer dominating the bulk characteristics, the discussion in section 4.4 has been revised to read:*

*"If primary production were significantly affecting any of the ice horizons, we would expect points on a graph of nTA vs. nTIC (Fig. 8) to fall along the photosynthesis/respiration line. Although statistically significant linear trends (p < 0.01) are apparent at both sites, and in all horizons, they do not follow the*

*photosynthesis line, even in the bottom ice horizon (Table 4). The slopes that we observed of nTA vs. nTIC all fell within the range of 0.50 – 1.03, with an overall slope of 0.65 at TFYI and 0.81 at POLY (Fig. 8). In their study of landfast sea ice in the Canadian Arctic Archipelago (Franklin Bay), Miller et al. (2011a) did see a few samples from May that fell along a photosynthesis line, but most followed a very similar slope to those presented in Fig. 8 and Table 4.*

*It may however be more instructive to consider the slope of the bottom horizon alone (Table 4), as the overall slopes are biased by the high number of samples collected at the surface and bottom, despite representing only a small portion of the overall ice volume. In the bottom horizon, we do see lower slope values (0.67 at TFYI, 0.50 at POLY), suggesting a shift towards the net respiration line, consistent with past studies that have found net heterotrophy in ice algae communities during the bloom (Campbell et al., 2017,2022; Rysgaard and Glud, 2004, Rysgaard et al. 2008). In the 2014 spring, Campbell et al. (2017) observed net heterotrophic conditions in the bottom ice of Dease Strait during a period of carbon accumulation, before switching to autotrophic conditions around 8-May. In a study near Resolute Bay, Brown et al. (2015) found substantial TIC drawdown near the beginning of an ice algae bloom, but that region experiences bottom ice Chl a value about 10 times higher than either of our sites (Leu et al., 2015). It is possible that the rate of ice algal production in Dease Strait is simply too low to have a noticeable impact on carbonate chemistry, or that primary production at our site was balanced or exceeded by respiration, highlighting that sea ice algae communities may have a complex role in carbon cycling in sea ice."*

*And portions of the discussion in section 4.5 have been revised, now reading:*

*"Of course, the TA:TIC ratio is not the only available evidence of ikaite precipitation in sea ice. There is microscopic evidence (e.g., Rysgaard et al., 2013), and crystals extracted from sea ice have been positively identified by x-ray diffraction (Dieckmann, et al. 2008). We did not have access to such techniques, but Figure 8 provides evidence that ikaite precipitation did occur in our study. At both sites, linear relationships between nTIC and nTA with slopes between 0.5 and 1.0 (Table 4), were observed through all horizons. These slopes are similar to those observed by Miller et al. (2011a), who hypothesized that such a relationship is the result of combined ikaite precipitation and respiration in the ice. Similar slope values were observed by Geilfus et al. (2012) and Brown et al. (2015) in brine. The differences we observed in slopes between horizons are likely the result of specific processes occurring near the surface (i.e. outgassing, see section 4.2) and the bottom (net respiration, see section 4.4) but with a strong tendency towards the carbonate precipitation/dissolution line in all horizons. Therefore, ikaite precipitation certainly seems to be occurring in our study area; our contention is simply that at this location (and apparently at several others) the amount of ikaite precipitation is not enough to result in a large fractionation of TA and TIC within the sea ice."*

*In addition, please see our response to Reviewer 1's comment RC1.01, which describes a minor change to the abstract.*

After addressing the above, as well as fixing the minor comments below, I would recommend this article for publication in The Cryosphere.
Cheers,
Eric Mortenson
Specific comments

**RC2.02** -84: I suggest adding underlined: …and under-ice seawater temperature and  currents…

*Change made*

**RC2.03** -104: I suggest adding underlined: …a function of under-ice seawater nutrient concentration and turbulence, …

*Change made*

**RC2.04** -259-260: Note that the respective magnitudes of tidal amplitudes and of tidally-induced currents are not necessarily related (e.g., currents can be quite strong at the mouth of an enclosed bay with a small opening, due to small amplitude changes in the enclosed bay)

*Good point, we revised this statement to improve clarity:*

*The increase in currents at POLY were likely due to a spring tide, but this was difficult to confirm with the Cambridge Bay tidal stage predictions (Fig. S1) where amplitudes are low due to the wide geometry of the bay.*

**RC2.05** -323: As mentioned just above this line, snow cover has a strong effect on light penetration, it would be nice to see a number for (or quantitative comparison to) Dalman's snow cover measurement.

*We have added:*

*"The distribution of ice algal biomass we observed was different from that reported in Dalman et al. (2019) who reported Chl a biomass averaging 9.3 mg m$^{-2}$ at the polynya site and 1.4 mg m$^{-2}$ at the thick ice site (c.f. Table 2) with an average snow thickness of 2.8 +/- 0.5 cm."*

**RC2.06** Table1/2: It would be nice to see these separated by date, in addition to the totals, at least for mean and standard deviation

*We created this table. It is rather large, and does not contribute significantly to the interpretation of data, although it does help show that the ice chemistry was quite stable between sampling dates. We will place it in the supplemental information. Note also that we will make all data on an open access data server available as per The Cryosphere's data policy, and will include citations to the database records in the revised manuscript.*

Technical corrections
**RC2.07** -Please be consistent using either DIC or TIC, but not both.

*Thank you, corrections have been made to TIC, except in a few instances in the introduction where DIC was the more appropriate term.*

**RC2.08** -Just a stylistic comment, why use "horizons", instead of strata or levels?

*Good question, we went back and forth on this ourselves. Our hope is that horizon is a well-understood term from geology/geography (e.g. soil horizons, geological horizons) that can be well understood by our audience.*

**RC2.09** -102-103, twice there are "(" with no ending parentheses, replace with commas maybe

*Thanks, we fixed this.*

**RC2.10** -Lines 119 and 125, Please spell Mortensen with "-son"

*Correction made, and our apologies!*

**RC2.11** -To me, it seems a bit clearer if the paragraph on all cores collected (lines 187-196) came before the paragraph on individual core carbon sampling (lines 177-186)

*We did find this section a little difficult to organize but prefer the current structure.*

**RC2.12** -Line 372: Nomura, Nomura should be Nomura, 2014

*Correction made*

**RC2.13** -Line 433: remove "to" at end of line

*Correction made*

**RC2.14** -Tables 1 and 2 appear to be identical, but with slightly different captions.

*Yes, sorry about that… Very bad "typo".  The revised manuscript has the correct table, and we have copied a version of this table in our response to Reviewer 1's comment RC1.08*

**RC2.15**-Fig. 1: I recommend changing the font color for POLY to white to improve visibility

*This change has been made, along with some other improvements for visibility. Please see our response to Reviewer 1's comment RC1.18.*

---

## Editor Decision (ED1)

[revised manuscript text omitted]